**Brief Communication**

# The chemotherapeutic drug CX-5461 is a potent mutagen in cultured human cells

Gene Ching Chiek Koh [1,2], Soraya Boushaki[2], Salome Jingchen Zhao [2], Andrew Marcel Pregnall [2], Firas Sadiyah [1,2], Cherif Badja [1,2], Yasin Memari[1,2], Ilias Georgakopoulos-Soares [3] & Serena Nik-Zainal [1,2] ✉

The chemotherapeutic agent CX-5461, or pidnarulex, has been fast-tracked by the United States Food and Drug Administration for early-stage clinical studies of *BRCA1*-, *BRCA2*- and *PALB2*-mutated cancers. It is under investigation in phase I and II trials. Here, we find that, although CX-5461 exhibits synthetic lethality in *BRCA1*-/*BRCA2*-deficient cells, it also causes extensive, nonselective, collateral mutagenesis in all three cell lines tested, to magnitudes that exceed known environmental carcinogens.

CX-5461 was initially characterized as a selective inhibitor of RNA polymerase I-dependent RNA synthesis, with application in hematological malignancies[1-3]. More recently, CX-5461 was reported to exhibit synthetic lethal properties, selectively killing *BRCA1*-/*BRCA2*-deficient cells[4,5]. This finding prompted several phase I dose escalation trials (ACTRN12613001061729, NCT02719977, NCT04890613)[6-8], alongside a joint selective therapeutics trial involving poly (ADP-ribose) polymerase inhibitor (PARPi) and CX-5461 (REPAIR, NCT05425862), in patients with relevant germline mutations (for example, *BRCA1*, *BRCA2* and/or *PALB2*). Proposed mechanisms underpinning therapeutic efficacy of CX-5461 include stabilizing G-quadruplexes (G4) and impeding topoisomerase II (TOP2) activity[9-11]. This could cause DNA damage, directly inducing mutations, yet the extent of its mutagenic potential has not been investigated in humans[12].

We exposed *BRCA1* and *BRCA2* knockouts (hereafter *ΔBRCA1* and *ΔBRCA2)* in hTERT-immortalized *TP53*-null retinal pigment epithelial 1 (RPE1) cells to pharmacologically relevant doses of CX-5461 and two other compounds with related mechanisms of action: etoposide (ETO, a TOP2 poison) and pyridostatin (PDS, a G4 ligand capable of trapping TOP2 on DNA[13]) (Fig. 1a and Supplementary Table 1). Following repeated cycles of treatment and recovery over ~35 days, mimicking a clinical dosing schedule, two to four single-cell daughter subclones were derived per genotype per treatment for whole-genome sequencing (WGS). De novo mutations acquired due to drug exposure were identified in each daughter subclone (Supplementary Table 2).

Surprisingly, CX-5461-treated clones showed high levels of mutagenesis of substitutions (SBS), double substitutions (DBS) and small insertions and deletions (indels) across all *ΔBRCA1*, *ΔBRCA2*

and control clones, compared with their untreated counterparts and other treatments (Fig. 1b). CX-5461 substitution burdens were 10–13 times greater than all other treatments, irrespective of genotype, exhibiting between 22,000 and 31,000 absolute mutations, rivaling burdens observed in human cancers[14,15]. Furthermore, CX-5461 showed a striking substitution pattern (or mutational signature) previously unreported—hitherto referred to as SBS-CX-5461. This signature is dominated by T>A and T>C mutations enriched at ATA, ATG trinucleotides (mutated base underlined) (Fig. 1c and Extended Data Fig. 1a,b). All genotypes showed near identical SBS-CX-5461 (cosine similarity, 0.99), although subtle differences were noticeable between *ΔBRCA1* and *ΔBRCA2* (Extended Data Fig. 1b,c). We also identified the substitution signature previously reported as SBS3 (associated with homologous recombination deficiency (HRd)) in all *ΔBRCA1* and *ΔBRCA2* cells (Fig. 1c,d and Extended Data Fig. 1b–d).

CX-5461 also generated a DBS pattern marked by AT>CA/GA/TA and TG>AT/CT/GT (DBS-CX-5461) (Fig. 1c). In silico permutations to ascertain the probability of coincidental double substitutions (arising from high substitution load) showed that the observed DBS-CX-5461 pattern differed from the predicted chance-related DBS pattern (Extended Data Fig. 1e). Manual assessment confirmed that these double substitutions were in cis, corroborating DBS-CX-5461 as a legitimate DBS signature. The burden of double substitutions was tenfold higher in treated cells compared with untreated counterparts (Fig. 1b). We also identified an indel signature (InD) for CX-5461 (InD-CX-5461) (Fig. 1b–d). The indel pattern was dominated by 1 bp T deletions at ATA and ATG (motifs enriched in SBS-CX-5461), 1 bp T insertions at $[T_{0-1}]A$, 2–4 bp duplications at nonrepetitive sequences,

[1]Department of Oncology, Early Cancer Institute, University of Cambridge, Cambridge, UK. [2]Academic Department of Medical Genetics, School of Clinical Medicine, University of Cambridge, Cambridge, UK. [3]Department of Biochemistry and Molecular Biology, Institute for Personalized Medicine, The Pennsylvania State University College of Medicine, Hershey, PA, USA. ✉e-mail: sn206@cam.ac.uk

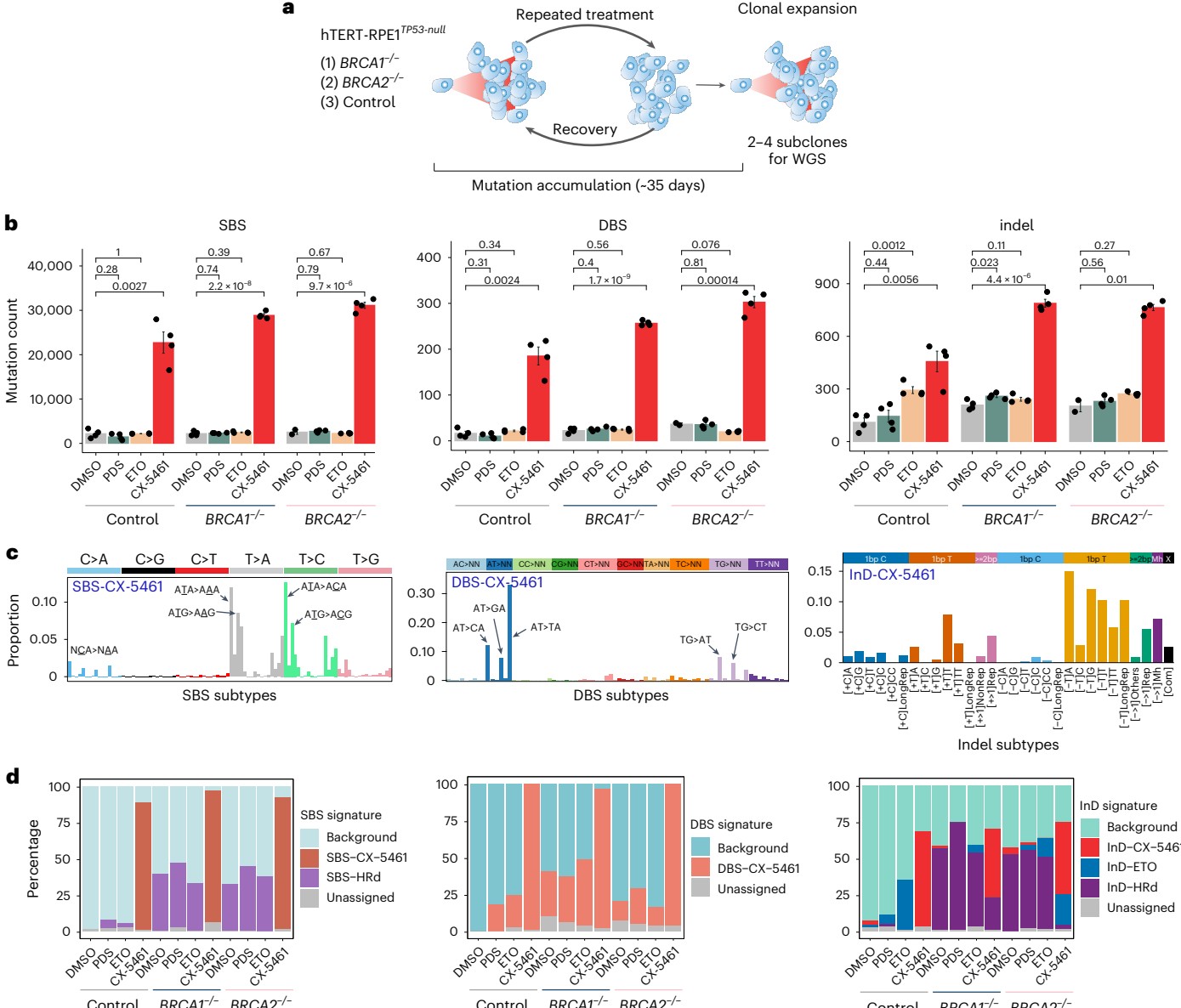

**Fig. 1 | CX-5461 induces heavy mutagenesis, leaving distinctive mutational signatures in hTERT-immortalized RPE1 cells. a**, Mutation accumulation experiment in which isogenic RPE1-*BRCA1*[−/−], -*BRCA2*[−/−] and control cells were treated with compounds of interest (PDS, ETO, CX-5641) or vehicle control (DMSO) repeatedly, over ~35 days and allowed to recover. Subsequently, two to four independent subclones were isolated per treatment per genotype and expanded for WGS. **b**, De novo mutation counts. Bars are mean ± s.e.m., *n* = 2–4

independent subclones per treatment per genotype (Supplementary Table 2). Two-tailed Student's *t* test was used to calculate *P* values. **c**, SBS, DBS and small indel signatures (InD) of CX-5461. **d**, Prevalence of signatures across different treatments and genotypes. SBS-HRd (substitution signature previously reported as SBS3 (associated with HRd) was averaged from SBS-BRCA1 and SBS-BRCA2 (Extended Data Fig. 1b); InD-HRd was averaged from InD-BRCA1 and InD-BRCA2 (Extended Data Fig. 1f).

and >5 bp deletions at microhomologies. The 2–4 bp duplications and microhomology-mediated deletions were redolent of the ETO indel signature (InD-ETO) (Fig. 1c,d and Extended Data Fig. 1f), supporting the mechanistic proposition that CX-5461 might exert TOP2 inhibitory effects[9–11] (Fig. 1c,d and Extended Data Fig. 1f). Slight differences in InD-CX-5461 were discernible between *ΔBRCA1* and *ΔBRCA2* (Extended Data Fig. 1f,g). The indel burden was nearly four times higher in CX-5461-exposed cells than in untreated controls (Fig. 1b).

To investigate mutational mechanisms underpinning CX-5461 mutagenesis, we inspected how they were distributed throughout the genome. We found notable depletion of SBS-CX-5461 mutations at predicted G4s (Fig. 2a), compatible with a formed secondary structure protecting G4 sequences. Intriguingly, we observed a conspicuous

periodicity immediately flanking G4s at a scale of ~200 bp, in keeping with the periodicity reported of nucleosomes. It is thus possible that CX-5461 initially fosters G4 stabilization, which subsequently promotes nucleosome reshuffling around these stabilized G4s, rendering linker regions between nucleosomes more susceptible to CX-5461-related damage. To corroborate this, we investigated CX-5461 mutation distribution relative to sites of stable nucleosome occupancy (Fig. 2b). Our result substantiates a prominent nucleosome-related periodicity, with enrichment at exposed linker regions amidst nucleosome cores. Moreover, CX-5461 mutations were evidently enriched in AT-rich, open chromatin regions, unaffected by replication timing (Fig. 2c), befitting rapid and substantial DNA damage engendered by CX-5461, primarily at open, exposed AT-rich regions.

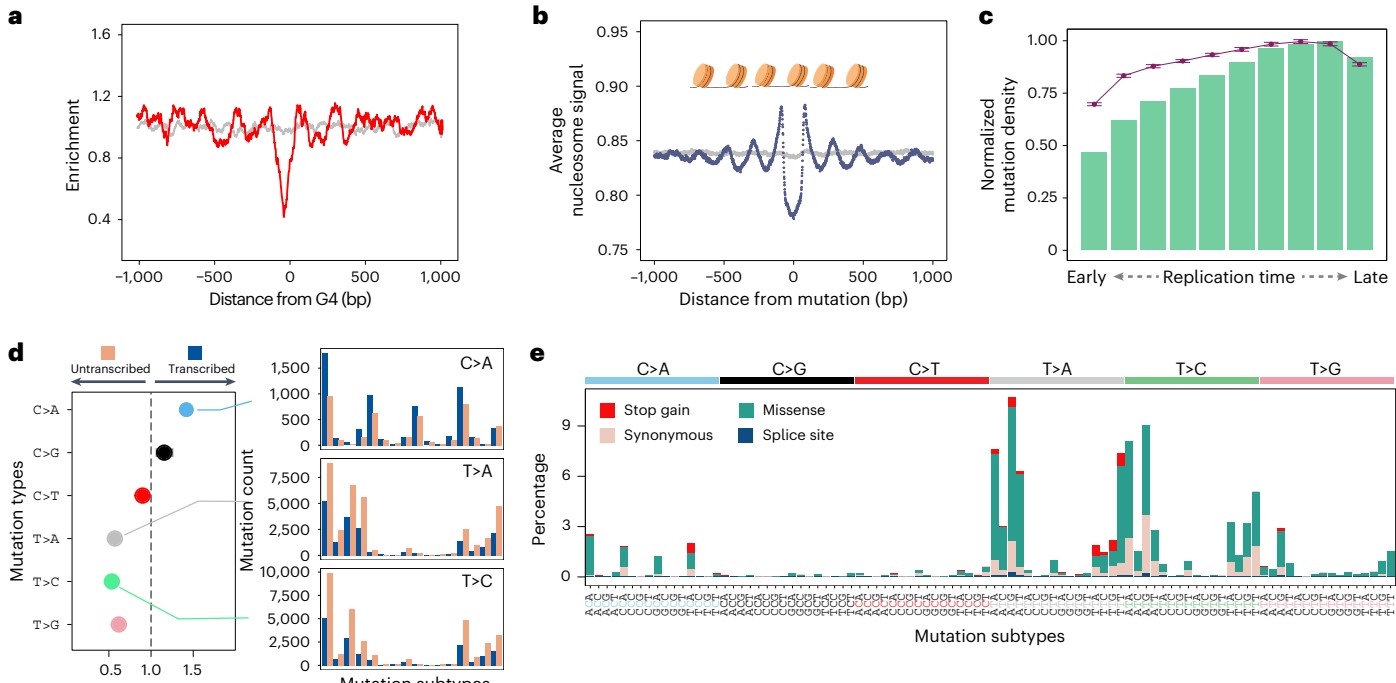

**Fig. 2 | Diverse mechanisms underpin synthetic lethality of CX-5461 and its mutagenicity. a**, Depletion of CX-5461 mutations at and around predicted G4s. The gray line represents simulated mutations controlling for trinucleotide context and proximity to original mutation (within 10 kb); the red line shows depletion of actual mutations. **b**, Nucleosome density for SBS-CX-5461 mutations. the gray line shows the distribution predicted by simulation if mutations were distributed randomly; the dark blue line shows average nucleosome signal for real mutations. **c**, Normalized SBS-CX-5461 mutations across cell cycle, from early to late replication timing regions (separated into deciles, left to right). Purple dots and error bars represent the mean ± s.d. of predicted SBS-CX-5461 mutations from $n = 100$ bootstrapped replicates. Green bars represent the distribution of observed substitution mutations from $n = 4$ subclones treated with CX-5461. **d**, Transcriptional strand asymmetry of SBS-CX-5461 mutations (Supplementary Table 3). **e**, Percentage of possible stop gain, missense, synonymous and splice site mutations based on SBS-CX-5461 mutation contexts against COSMIC Cancer Gene Census Tier 1 and 2 cancer genes (Supplementary Table 5).

Next, we sought clues of DNA repair activity involved in addressing CX-5461 damage. We noted a strong strand asymmetry of T>A, T>C, T>G and G>T (or C>A) mutations towards the untranscribed strand (Fig. 2d). This aligns with the activity of transcription-coupled repair preferentially repairing damage on the transcribed strand. We did not observe asymmetry in the mutagenesis of replicative strands (Supplementary Table 3). Taken together, our analyses suggest that, whereas the cytotoxic effects of CX-5461 may be driven through TOP2 poisoning caused by G4 stabilization, its mutagenic effects likely stem from alternative mechanisms—plausibly bulky, DNA-deforming adducts occurring at exposed, AT-rich genomic regions in a sudden and catastrophic manner, accounting for the conspicuous topographical distributions noted above.

Finally, we checked that CX-5461 exhibited synthetic lethality in *ΔBRCA1* and *ΔBRCA2* cells, and not in unedited controls. We confirmed selective synthetic lethality of *ΔBRCA1, ΔBRCA2* and *ΔLIG4* cells for CX-5461 (ref. 13) (Extended Data Fig. 2). Critically, this implies that, although CX-5461-induced lethality is selective towards *BRCA1-, BRCA2-*deficient cells, mutagenesis is not; healthy and normal cells that are exposed to CX-5461 may be mutagenized.

To test whether CX-5461-induced mutational signatures are a universal DNA damage phenomenon observable across other cell types and doses, we applied an acute 24-h exposure of 0.1 μM CX-5461 to *TP53-null* HAP1 cells—a near-haploid line derived from a hematological cancer. WGS of CX-5461-exposed HAP1 subclones revealed the presence of SBS-CX-5461, DBS-CX-5461 and InD-CX-5461. The SBS and DBS signatures bore very high resemblance to the signatures derived in RPE1 cells (cosine similarities of 0.944 and 0.887, respectively). The indel signature had a lower similarity (0.496) because of the generally lower indel rate and a known strong background indel signature in

HAP1 (ref. 16) (Extended Data Fig. 3a). This highlights how a singular dose of CX-5461 is potent enough to generate marked mutagenesis in an alternative cell model. Further, we asked whether a very short exposure to CX-5461 (of only 2 h) could generate mutations. We used duplex sequencing[17] to seek ultralow-frequency variants within bulk cell populations following exposure in yet another model—human induced pluripotent stem cells (hiPSCs) (Supplementary Tables 1 and 4). We contrasted CX-5461 with an established carcinogenic environmental compound, benzo(a)pyrene (BaP)—a polycyclic aromatic hydrocarbon found in tobacco smoke—and other chemotherapeutics (cisplatin, ETO and PDS). Even with a single, ultrashort (2 h) and low (0.1 μM) dose, CX-5461 yielded ~1.5 times the number of mutations of BaP and ~2.6 times over untreated control in hiPSCs, underscoring how potently this compound incurs DNA damage (Extended Data Fig. 3b–d). Indeed, our results suggest that it is more mutagenic than known environmental agents[18,19], including those associated with cancer risk.

To compare potential impact directly within relevant clinical contexts, we contrasted the mutagenicity of CX-5461 to cisplatin and the PARPi Olaparib—alternative therapeutic agents used in *BRCA1-/BRCA2-*deficient breast and ovarian cancer patients. While PARPi does not generate mutational signatures, cisplatin produces SBS, DBS and indel mutational signatures[18]. We calculated a mutagenicity index (MI), which considers overall mutation burden. CX-5461 had MI values of ~6.8, 7.1 and 2.1 for SBS, DBS and indels, respectively. By contrast, cisplatin had MI values of 0.6, 11.6 and 1 (ref. 18). Thus, compared with platinum, CX-5461 is nearly over ten times more mutagenic for SBS and around two times more mutagenic for indels. Finally, we calculated the 'damage potential' of SBS-CX-5461, that is,

the likelihood of incurring new driver events should the same pattern occur in coding sequences with consequential amino acid changes. Although we did not observe the acquisition of driver mutations in these short-term experiments, compared with background mutagenesis, SBS-CX-5461 was estimated to exhibit 1.15 higher odds of causing a stop gain mutation in genes causally implicated in cancer (COSMIC Tier 1/2 cancer genes) (Supplementary Table 5). This is much higher than signatures associated with common environmental exposures (for example, smoking-SBS4, 0.86; ultraviolet radiation-SBS7a, 0.15; Platinum-SBS31, 0.70, respectively).

Cancer is a multifactorial disease influenced by a multitude of genetic and environmental factors. Unsurprisingly, environmental exposures like tobacco or ultraviolet radiation, known for their mutagenic potential, have been linked to increased cancer risk, leading to public health initiatives to minimize exposure. Many chemotherapeutic agents commonly target DNA replication and/or induce DNA damage to provoke cell death. The possibility of lasting DNA damage, however, is an outcome that must be balanced against the chemotherapeutic benefits offered by these treatments in combating cancer. This is the first time we have encountered a chemotherapeutic agent with such a pronounced mutational phenotype, surpassing the effects of all other environmental mutagens and chemotherapeutics analyzed in a systematic screen[18].

Notably, dramatic CX-5461 mutagenesis was observed across three distinct human cellular models and among all genetic backgrounds, including normal control cells. This contrasts with its anticipated physiological impact, which is believed to be selectively lethal only for *BRCA1-/BRCA2*-deficient cells. Consequently, while CX-5461 may not eliminate normal cells, its profoundly mutagenic outcomes likely impact them. This mutagenic effect does not limit itself to HRd cells, thereby carrying a detrimental implication that could potentially contribute to future cancer risk, although this will need to be fully explored. Given these findings and the roll-out of this drug into clinical trials, we urge the community to reconsider the use of CX-5461 in human patients until additional evidence is obtained to evaluate its potential for causing cancer. We acknowledge that our results are from in vitro systems and that any in vivo mutagenic effects will require investigation under ethically acceptable conditions. Moving forward, we suggest that mutagenicity of new drugs needs to be comprehensively evaluated before human trials.

## Online content

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

## Methods

### Cell culture

The generation of hTERT-RPE1 *ΔTP53, ΔBRCA1* and *ΔBRCA2* cells has been described elsewhere[20,21]. They were gifts from M. Tarsounas (University of Oxford) and cultivated in Dulbecco's Modified Eagle Medium/Ham's Nutrient Mixture F-12 (Gibco/Thermo Fisher Scientific). HAP1 *ΔTP53* cells[22] were obtained from J. Loizou (CeMM, Austria) and maintained in Iscove's Modified Dulbecco's Medium with GlutaMAX supplement (Gibco/Thermo Fisher Scientific). Media for HAP1, hTERT-RPE1 and their derivatives were supplemented with 10% fetal bovine serum.

The hiPSC line was derived at the Wellcome Trust Sanger Institute and has been published[18]. The use of this cell line model was approved by Proportionate Review Subcommittee of the National Research Ethics Committee North West–Liverpool Central under the project 'Exploring the biological processes underlying mutational signatures identified in induced pluripotent stem cell lines (iPSCs) that have been genetically modified or exposed to mutagens' (ref: 14.NW.0129). It is a long-standing iPSC line originally isolated from a patient with α-1-antitrypsin deficiency, for which one of the alleles was corrected. The cell line is karyotypically stable and does not carry any known driver mutations. It does, however, carry a balanced translocation between chromosomes 6 and 8. Stem cell culture reagents were sourced from StemCell Technologies unless otherwise indicated. Cells were routinely maintained on Vitronectin XF-coated plates (10–15 µg ml$^{-1}$) in Essential 8 Basal Medium (Gibco/Thermo Fisher Scientific). The medium was changed daily and cells were passaged every 4–8 days depending on the confluence of the plates using 0.5 mM EDTA. All cell lines were maintained at 37 °C and 5% $CO_2$ in a humidified incubator.

### Drug sensitivity assay

The Celltiter-Glo v.2.0 assay (Promega, catalog no. G9243) was used to assess cell viability following the applied drug treatment. The assay determines the number of viable cells in culture based on the quantitation of ATP present, which serves as a proxy for the number of metabolically active cells. A total of 300 cells per well were seeded in 96-well plates in a volume of 100 µl medium; 24 h later, cells were treated with increasing concentrations of respective compounds in triplicate. Cells were maintained in drug-containing medium for 6 days and luminescence signals were quantified following the manufacturer's instructions. The surviving fraction of drug-treated cells was normalized to values from respective solvent-treated controls. Compound half-maximum inhibitory concentration and statistics were calculated using GraphPad prism software (GraphPad v.9.5.1).

### Drug treatment

Cells were treated with each compound at a concentration that results in 40–60% cytotoxicity, in parallel with cells treated with dimethylsulfoxide (DMSO) solvent control. Drug exposure frequencies, dosages and duration are detailed in Supplementary Table 1.

### Mutation accumulation and WGS

Cell lines were maintained in culture, with or without treatment, for around 35 days (about 30 cell doublings) to allow for mutation accumulation. Following that, a second round of single-cell limiting dilution was performed to isolate two to four daughter subclones per experimental arm for WGS, providing a bottleneck to capture mutations that had occurred since the isolation of the initial drug-treated or untreated parental clones.

Genomic DNA was isolated from all pelleted cell lines using Quick-DNA Miniprep Plus Kit (ZymoResearch) following the manufacturer's protocol. WGS libraries were prepared and sequenced with a paired-end 150 bp configuration on an Illumina NovaSeq 6000 platform by Novogene, aiming for an average genome-wide sequencing depth of 25× per sample.

### Somatic variant calling

WGS short reads were aligned to GRCh38/hg38 using BWA-MEM v.0.7.17-r1188. Quality control and bioinformatic analysis of the WGS data was performed using CaVEMan[23] (v.1.13.15) for SBS and DBS, Pindel[24,25] (v.3.2.0) for indels, BRASS (https://github.com/cancerit/BRASS, v.6.2.1) for rearrangements and ASCAT (NGS) (https://github.com/cancerit/ascatNgs, v.4.2.1) for copy number variations. Postprocessing filters were applied to improve the specificity of mutation calling. Specifically, for single nucleotide variant calls by CaVEMan[23], we used CaVEMan filters CLPM = 0 and ASMD ≥ 140. To reduce false positive calls by Pindel[24], we used Pindel filters QUAL ≥ 250 and REP < 10. Rearrangements were not assessed as they were too few to be informative. Variant allele fraction (VAF) distribution for each subclone was examined, and those with an average VAF < 0.4 were designated as polyclonal and subsequently excluded from all quantitative analyses (that is, estimation of mutation density and mutation burden). A filter for variant allele frequency (>0.2) was applied to substitutions and indels. De novo substitutions and indels in subclones were obtained by subtracting from respective parental clone whenever available, or by removing mutations shared among subclones. De novo mutation counts are provided in Supplementary Table 2.

### Mutational signature analysis of experimental samples

Experimental mutational signatures were derived using the published framework (https://github.com/xqzou/COMSIG_KO) based on cosine similarity, profile bootstrapping and background subtraction[26,27]. Briefly, we (1) determined the background mutational signature in unedited/untreated control by aggregating the unedited and untreated subclone mutational profiles, then (2) assessed the difference/s between the mutational profiles of the edited/treated clones and the controls using cosine similarity. Specifically, we first evaluated the similarity of mutational profiles between the untreated control and each subclone. We calculated the cosine similarity between each bootstrapped control sample and the aggregated background control mutational signature from (1) (means and s.d. values). A cosine similarity close to 1.0 indicates that the mutation profile of the bootstrapped sample is near identical to the control signature. Cosine similarities could thus be considered across a range of mutation burdens (green, pink and blue line for SBS, DBS and indel, respectively, in Extended Data Fig. 1a). Next, we calculated cosine similarities between edited/treated subclone profiles and control (colored shapes in Extended Data Fig. 1a). An edit or a treatment that does not fall within the expected distribution of cosine similarities implies a mutation profile distinct from controls (that is, the perturbation generated a signature). If an edit or a treatment generates a signature, we (3) removed background mutation profile from the mutation profile of edited/treated clones. Experimentally derived signatures were compared with published reference signatures[15] using signature.tools. lib from https://rdrr.io/github/Nik-Zainal-Group/signature.tools.lib/.

Although CX-5461-treated subclones did show a slight increase in rearrangement counts and chromosomal copy number aberrations compared with their untreated counterparts, the counts were too low and insufficiently powered to draw any conclusions (Supplementary Table 2).

### G4 enrichment analysis

We used the genome-wide G4 maps for the human genome from the consensus G4 motif (G≥3N1–7G≥3N1–7G≥3N1–7G≥3)[28]. We generated a 2-kb window centered at the somatic mutations and calculated the distribution of G4s. The fold enrichment of G4 relative to somatic mutations was calculated as the ratio of the number of G4 occurrences at each position, over the median number of occurrences across the whole window (enrichment = score at position/mean score across positions).

### Nucleosome positioning analysis

Micrococcal nuclease sequencing data for the K562 cell line was obtained from the ENCODE project[29]. To assess the relationships

between SBS-CX-5461 mutations and nucleosome occupancy, we created a window of 2 kb centered around each mutation in CX-5461-treated samples and obtained the nucleosome density signal observed within the 2-kb window. We calculated the sum of the signal observed (SUM) across the window for all the mutations within SBS-CX-5461, and the number of mutations (NUM) contributing to the signature. The average signal ($y$ axis) is the SUM/NUM for every position within the 2-kb window.

Mutations contributing to a given signature are scattered across different genomic locations, often numbering in the thousands or even tens or hundreds of thousands. If these mutations were independent of nucleosome positioning, the aggregated data would exhibit a flat line. However, if mutations within a specific signature showed a tendency to occur at core sequences, a pronounced peak in the nucleosome signal would be observed at the mutation center. Conversely, if mutations were more prevalent in linker sequences, a noticeable trough would emerge in the nucleosome signal.

### TwinStrand DuplexSeq

Duplex sequencing[17] was carried out following the manufacturer's protocol. Briefly, genomic DNA (1,000 ng) of treated and untreated cells was fragmented enzymatically and paired-end Illumina sequencing libraries were created using the TwinStrand Duplex sequencing mutagenesis kits for human panels. The protocol comprises several key steps: end-repair, A-tailing, ligation of DuplexSeq adapters and treatment with a conditioning enzyme cocktail to eliminate chemically damaged bases before PCR amplification using unique dual index-containing primers. Following template indexing and amplification, two consecutive rounds of hybrid selection for mutagenesis target enrichment were performed using a pool of biotinylated oligonucleotides. The enriched samples were washed and a final PCR step was performed to add on the P5/P7 primers. Subsequently, all resulting DuplexSeq libraries were quantified, pooled and sequenced on an Illumina NovaSeq 6000 S2 flow cell, with 150-bp paired-end specification to achieve a target of around 1.2 billion informative duplex bases per sample. The sequencing was performed using vendor-supplied reagents and v.1.0 chemistry.

Analyses were performed using the TwinStrand DuplexSeq Mutagenesis App, hosted on DNAnexus. The Mutagenesis App performed error-correction and generated Duplex Consensus alignment and variant calls for both germline and ultrarare somatic variants. Only variants with variant allele frequency <0.01 were considered to be the result of mutagenesis (that is, mutation) and included for subsequent mutation burden and signature analysis (Supplementary Table 4).

### Damage potential analysis

Damage potential analysis was done using MutationalPatterns (https://github.com/UMCUGenetics/MutationalPatterns)[30]. Briefly, the method involves quantifying the ratio of different mutation types (that is, 'stop gain,' 'missense,' 'synonymous mutations' and 'splice site mutations') within each signature. To provide a standardized measure, these ratios were normalized by comparing them with the ratios observed in a completely random 'flat' signature. A normalized ratio of 2 for 'stop gain' mutations, for example, indicates that a signature is twice as likely to cause 'stop gain' mutations compared with the random baseline. The calculation of these ratios involves multiplying the number of possible mutations per context by the signature contribution per context and summing over all contexts (Supplementary Table 5). Additionally, the method computes the blosum62 score for mismatches, indicating the dissimilarity between amino acids. A lower score suggests greater dissimilarity and a higher likelihood of detrimental effects. Normalized blosum62 scores are also determined by subtracting the score of the 'flat' signature from the base blosum62 scores.

### Statistics and reproducibility

All comparisons were between biologically independent samples. No statistical method was used to predetermine sample size. No data were excluded from the analyses. The experiments were not randomized. The investigators were not blinded to allocation during experiments and outcome assessment. Further details are provided in the Reporting Summary.

### Reporting summary

Further information on research design is available in the Nature Portfolio Reporting Summary linked to this article.

### Data availability

Raw sequence files from hTERT-RPE1 and HAP1 mutation accumulation experiments are deposited at the European Genome-Phenome Archive with dataset ID EGAD50000000036. Mutation calls have been deposited at Mendeley and can be accessed via https://doi.org/10.17632/d58cv549v6.1. Downstream data are provided in the Supplementary Tables. All cell line models cells can be requested directly from the corresponding author. Curated data are available for general browsing from Signal (https://signal.mutationalsignatures.com) upon publication.

### Code availability

No custom code or software was generated as part of the study. Details of all software packages used for data processing and/or analysis may be found in Methods.

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

### Acknowledgements

This work was funded by a Cancer Research UK (CRUK) Advanced Clinician Scientist Award (C60100/A23916) to S.N.-Z., a Dr. Josef Steiner Cancer Research Award 2019 to S.N.-Z., a Basser Gray Prime Award 2020 to S.N.-Z., a CRUK Pioneer Award (C60100/A23433) to

S.N.-Z., a CRUK Grand Challenge Award (C60100/A25274) to S.N.-Z., a CRUK Early Detection Project Award (C60100/A27815) to S.N.-Z. and a National Institute of Health Research (NIHR) Research Professorship (NIHR301627) to S.N.-Z. This work was also supported by the NIHR Cambridge Biomedical Research Centre (BRC-1215-20014; S.N.-Z). The funders had no role in study design, data collection and analysis, decision to publish or preparation of the manuscript.

## Author contributions

G.C.C.K. and S.N.-Z. conceived the project and designed the experiments. G.C.C.K., S.B., S.J.Z. and C.B. performed gene-editing, drug treatment, mutation accumulation and sequencing experiments. G.C.C.K., S.B., A.M.P., F.S., Y.M. and I.G.-S. implemented computational analyses. S.N.-Z. supervised the work. Data interpretation and write-up were provided by G.C.C.K. and S.N.-Z., with input from all the other authors, who had the opportunity to edit the manuscript and approved of the final submitted version.

## Competing interests

S.N.-Z. holds patents or has submitted applications on clinical algorithms of mutational signatures: MMRDetect (PCT/EP2022/057387), HRDetect (PCT/EP2017/060294), clinical use of signatures (PCT/EP2017/060289), rearrangement signature methods (PCT/EP2017/060279), clinical predictor (PCT/EP2017/060298) and hotspots for chromosomal rearrangements (PCT/EP2017/060298). Two further patent filings have been made recently (numbers are pending). All other authors declare no competing interests.

## Additional information

**Extended data** is available for this paper at https://doi.org/10.1038/s41588-023-01602-9.

**Correspondence and requests for materials** should be addressed to Serena Nik-Zainal.

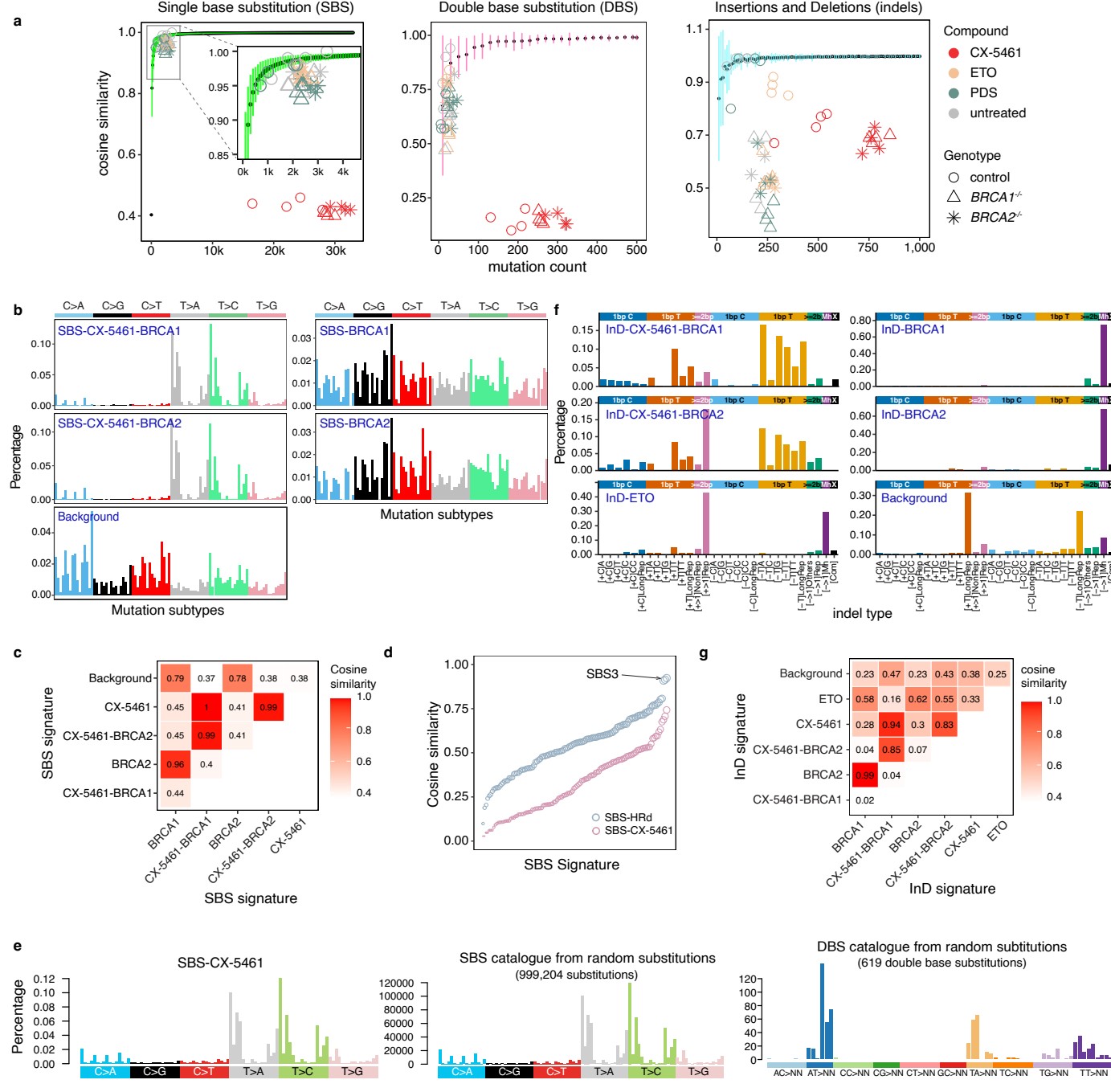

**Extended Data Fig. 1 | Mutational signatures of CX-5461 in isogenic hTERT-RPE1 cells. a**. Distinguishing *de novo* mutational profiles of experimental subclones from controls. Light green, pink, and blue error bars (left to right) depict the mean ± 3SD of cosine similarities between *n* = 100 bootstrapped control profiles and the control mutational profile aggregated from *n* = 4 DMSO-treated control subclones, of respective mutation types with increasing mutation counts. The *x* axis displays the mutation counts for respective mutation classes. See Methods for details. **b**. Single base substitution (SBS) signatures of gene knockouts and CX-5461 in different knockout backgrounds. Background signature was derived from untreated RPE1 cells. **c**. Heatmap

showing cosine similarities between experimental SBS signatures. **d**. Cosine similarities comparing SBS-CX-5461 and SBS-HRd to reference SBS signatures. **e**. *in silico* permutation to assess whether DBS-CX-5461 is a chance occurrence due to high mutation burden given SBS-CX-5461 pattern. DBS, double base substitution. **f**. Small insertion and deletion signatures associated with homologous recombination deficiency (HRd), etoposide (ETO), and CX-5461 exposure. InD-BRCA1 and InD-BRCA2 were identical (cosine similarity, 0.99), and hence averaged as InD-HRd. Background signature was derived from untreated RPE1 cells. **g**. Heatmap showing cosine similarities between experimental indel signatures (InDs).

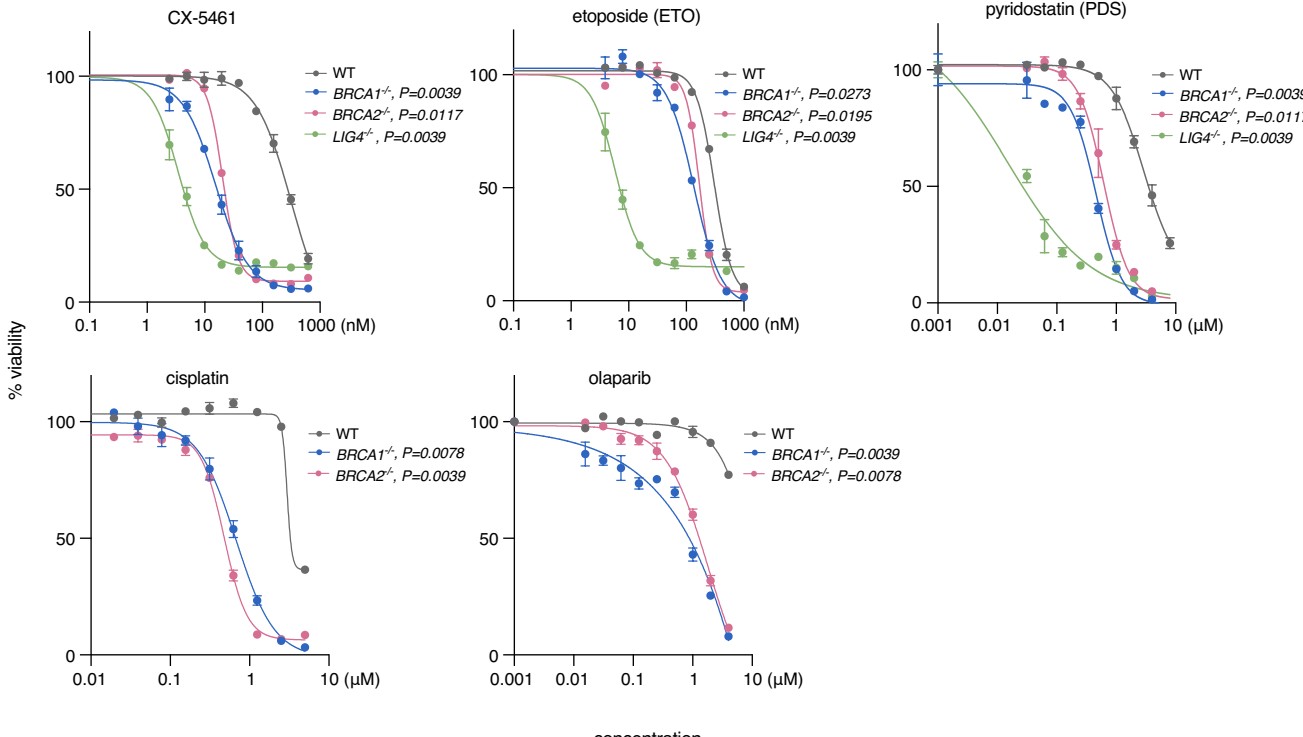

**Extended Data Fig. 2 | Drug sensitivity profiling by CellTiter-Glo cell viability assay.** Drug sensitivity profiling of isogenic RPE1-*BRCA1*⁻/⁻, *BRCA2*⁻/⁻, and control cells to CX-5461, topoisomerase II poison, etoposide (ETO), and G-quadruplex stabilising compound, pyridostatin (PDS) confirmed synthetic lethality of CX-5461 in RPE1-*BRCA1*⁻/⁻, *BRCA2*⁻/⁻ cells. RPE1-*LIG4*⁻/⁻ was used as a positive control.

Cells were also profiled against two other therapeutic agents commonly used for the treatment of *BRCA1/2*-mutated cancers, cisplatin, and olaparib. Data are mean ± standard errors (error bars), *n* = 3 independent biological replicates. All comparisons were made against WT. Two-tailed Wilcoxon signed-rank test was used to calculate *P* values.

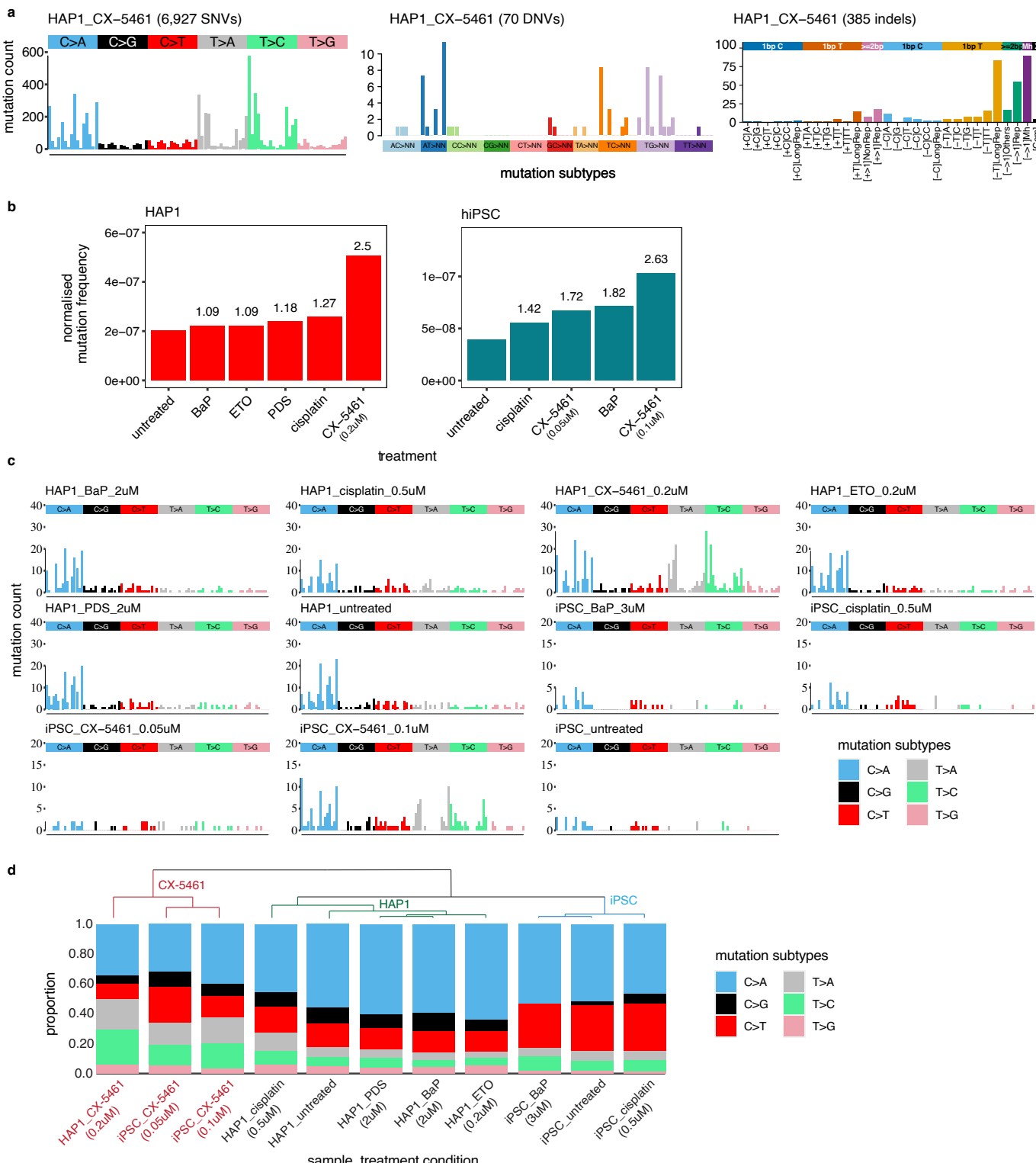

**Extended Data Fig. 3 | Validation of CX-5461 signatures in alternative cellular models. a**. Aggregated whole-genome mutational profiles of CX-5461-treated HAP1 subclones (*n* = 2). **b**. Mutation frequencies normalized by the total duplex bases per sample across different compounds in HAP1 (left) and human induced pluripotent stem cells (hiPSC) (right) (*n* = 1 per treatment arm). Mutation frequency fold-increases were calculated against respective untreated control (bar top). BaP, benzo(*a*)pyrene; ETO, etoposide; PDS, pyridostatin. **c**. Trinucleotide spectrum plots for treated bulk cells by duplex sequencing. **d**. Unsupervised hierarchical clustering of mutational spectra (six mutation types) collapsed from **c**. using (*1-cosine similarity*) as distance matrix. BaP, benzo(*a*) pyrene; Cis, cisplatin; ETO, etoposide; PDS, pyridostatin.

# Reporting Summary

## Statistics

For all statistical analyses, confirm that the following items are present in the figure legend, table legend, main text, or Methods section.

| n/a | Confirmed | |
|---|---|---|
| ☐ | ☒ | The exact sample size (*n*) for each experimental group/condition, given as a discrete number and unit of measurement |
| ☐ | ☒ | A statement on whether measurements were taken from distinct samples or whether the same sample was measured repeatedly |
| ☐ | ☒ | The statistical test(s) used AND whether they are one- or two-sided<br>*Only common tests should be described solely by name; describe more complex techniques in the Methods section.* |
| ☒ | ☐ | A description of all covariates tested |
| ☒ | ☐ | A description of any assumptions or corrections, such as tests of normality and adjustment for multiple comparisons |
| ☐ | ☒ | A full description of the statistical parameters including central tendency (e.g. means) or other basic estimates (e.g. regression coefficient) AND variation (e.g. standard deviation) or associated estimates of uncertainty (e.g. confidence intervals) |
| ☐ | ☒ | For null hypothesis testing, the test statistic (e.g. *F*, *t*, *r*) with confidence intervals, effect sizes, degrees of freedom and *P* value noted<br>*Give P values as exact values whenever suitable.* |
| ☒ | ☐ | For Bayesian analysis, information on the choice of priors and Markov chain Monte Carlo settings |
| ☒ | ☐ | For hierarchical and complex designs, identification of the appropriate level for tests and full reporting of outcomes |
| ☒ | ☐ | Estimates of effect sizes (e.g. Cohen's *d*, Pearson's *r*), indicating how they were calculated |

*Our web collection on statistics for biologists contains articles on many of the points above.*

## Software and code

Policy information about availability of computer code

| | |
|---|---|
| Data collection | We performed whole genome sequencing and TwinStrand duplex sequencing of all experimental control and treatment samples on Illumina Novaseq 6000 platform generating 150 base pair paired-end reads.<br><br>Mutagen exposure WGS data of human induced pluripotent stem cells (used for calculating mutagenicity index) were published and were accessed via https://data.mendeley.com/datasets/m7r4msjb4c/2. |
| Data analysis | Experimental WGS short read data were aligned to the human reference genome GRCh38 assembly using "bwa mem 0.7.17-r1188". Quality control and bioinformatic analysis of the WGS data was performed using "CaVEMan v1.13.15 " for substitutions, "Pindel v3.2.0" for insertions/deletions, "Brass v6.2.1" for rearrangements, and "ASCAT (NGS) v4.2.1" for copy number variations. Experimental signature derivation was performed as described in doi: 10.1038/s43018-021-00200-0 and codes can be obtained from https://github.com/xqzou/COMSIG_KO and https://rdrr.io/github/Nik-Zainal-Group/signature.tools.lib/.<br><br>TwinStrand error-corrected duplex sequencing data analysis was performed using DuplexSeq Mutagenesis App v4.1.0 hosted on DNAnexus using default parameters, Pipeline ID: human-muta-v1.0.<br><br>Drug sensitivity profiling data were analysed with GraphPad version 9.5.1. |

For manuscripts utilizing custom algorithms or software that are central to the research but not yet described in published literature, software must be made available to editors and reviewers. We strongly encourage code deposition in a community repository (e.g. GitHub). See the Nature Portfolio guidelines for submitting code & software for further information.

# Data

Policy information about availability of data

All manuscripts must include a data availability statement. This statement should provide the following information, where applicable:

- Accession codes, unique identifiers, or web links for publicly available datasets
- A description of any restrictions on data availability
- For clinical datasets or third party data, please ensure that the statement adheres to our policy

Raw sequence files from hTERT-RPE1 and HAP1 mutation accumulation experiments are deposited at the European Genome-Phenome Archive with accession number EGAD50000000036. Mutation calls have been deposited at Mendeley and can be accessed via DOI: 10.17632/d58cv549v6.1. Downstream analyses data are provided in the Supplementary Tables. All cell line models cells can be requested directly from the corresponding author. Curated data are available for general browsing from Signal (https://signal.mutationalsignatures.com) upon publication.

# Research involving human participants, their data, or biological material

Policy information about studies with human participants or human data. See also policy information about sex, gender (identity/presentation), and sexual orientation and race, ethnicity and racism.

| Reporting on sex and gender | N/A |
|---|---|
| Reporting on race, ethnicity, or other socially relevant groupings | N/A |
| Population characteristics | N/A |
| Recruitment | N/A |
| Ethics oversight | N/A |

Note that full information on the approval of the study protocol must also be provided in the manuscript.

# Field-specific reporting

Please select the one below that is the best fit for your research. If you are not sure, read the appropriate sections before making your selection.

☒ Life sciences ☐ Behavioural & social sciences ☐ Ecological, evolutionary & environmental sciences

For a reference copy of the document with all sections, see nature.com/documents/nr-reporting-summary-flat.pdf

# Life sciences study design

All studies must disclose on these points even when the disclosure is negative.

| Sample size | From a statistical standpoint, this was an exploratory study, and there were no pre-defined hypothesis tests for which sample-size power calculations were appropriate. |
|---|---|
| Data exclusions | From a statistical perspective, this was an exploratory study, and there were no pre-defined hypothesis tests for which pre-defined data exclusion criteria would have been appropriate. Therefore, no data were excluded from by our algorithms. |
| Replication | Each gene edit/treatment arm had at least 2-4 daughter sub-clones as biological replicates. For TwinStrand duplex sequencing experiment, each treatment arm only has one duplex library being sequenced and analysed. Drug sensitivity profiling data were obtained from a minimum of three independent replicates. All attempts at replication were successful.<br><br>For few of the chemicals (for example, CX-5461), we replicated the analysis to identify CX-5461 associated signatures using different cell line models. For all the chemicals, each of the subclone genomes sequenced represents an independent data point and as such much of the paper explored the replication of signatures across the cell line collection. |
| Randomization | The question of allocation to experimental groups is not applicable to this study. No randomization was performed. All experimental samples were contrasted against respective isogenic unedited (WT) or untreated (DMSO solvent treated) controls. |
| Blinding | We applied the analysis algorithms to each and every perturbation (i.e., treatment) in the dataset in exactly the same way and without any prior expectations about the desired outcome of the analysis. Therefore, blinding was not required. |

# Reporting for specific materials, systems and methods

We require information from authors about some types of materials, experimental systems and methods used in many studies. Here, indicate whether each material, system or method listed is relevant to your study. If you are not sure if a list item applies to your research, read the appropriate section before selecting a response.

## Materials & experimental systems

| n/a | Involved in the study |
|-----|----------------------|
| ☒ | Antibodies |
| ☐ | ☒ Eukaryotic cell lines |
| ☒ | Palaeontology and archaeology |
| ☒ | Animals and other organisms |
| ☒ | Clinical data |
| ☒ | Dual use research of concern |
| ☒ | Plants |

## Methods

| n/a | Involved in the study |
|-----|----------------------|
| ☒ | ChIP-seq |
| ☒ | Flow cytometry |
| ☒ | MRI-based neuroimaging |

## Eukaryotic cell lines

Policy information about cell lines and Sex and Gender in Research

| | |
|---|---|
| Cell line source(s) | The original hTERT RPE-1 are hTERT-immortalized retinal pigment epithelial cells derived by transfecting the RPE-340 cell line with the pGRN145 hTERT-expressing plasmid. This is a near-diploid human cell line of female origin with a modal chromosome number of 46 that occurred in 90% of the cells counted. All hTERT-RPE1 cells that we used in this study was originally generated from doi: 10.1038/s41586-018-0291-z. They were gifts from M. Tarsounas, (Department of Oncology, University of Oxford, UK). HAP1 cell was obtained from J. Loizou (CeMM, Austria, DOI: 10.1038/s41467-017-01439-x). Human induced pluripotent stem cells were initially described in DOI: 10.1016/j.cell.2019.03.001. The human induced pluripotent stem cell (hiPSC) was derived at the Wellcome Trust Sanger Institute (Hinxton, UK). The use of this cell line model was approved by Proportionate Review Sub-committee of the National Research Ethics (NRES) Committee North West - Liverpool Central under the project "Exploring the biological processes underlying mutational signatures identified in induced pluripotent stem cell lines (iPSCs) that have been genetically modified or exposed to mutagens" (ref: 14.NW.0129). |
| Authentication | The cell lines were not authenticated in this study. However, we did have the whole genome-sequencing data and had matched SNP genotype profiles to confirm the cell line identities and their isogenicity. |
| Mycoplasma contamination | Stock cell lines were tested negative for mycoplasma contamination when banked and used the first time, but not tested again throughout the mutation accumulation, drug exposure, and single-cell subcloning steps. |
| Commonly misidentified lines (See ICLAC register) | Not applicable as none were used. |

