## [Peer Review File · Nature Genetics]

Peer Review Information

Manuscript Title: The chemotherapeutic drug CX-5461 is a potent mutagen in cultured human cells

Corresponding author name(s): Professor Serena Nik-Zainal

Reviewer Comments & Decisions:

Decision Letter, initial version:

19th Sep 2023

Dear Professor Nik-Zainal,

Your Brief Communication, "The chemotherapeutic CX-5461 is extremely mutagenic and may increase cancer risk" has now been seen by 3 referees. You will see from their comments below that while they find your work of interest, some important points are raised. We are interested in the possibility of publishing your study in Nature Genetics, but would like to consider your response to these concerns in the form of a revised manuscript before we make a final decision on publication.

We therefore invite you to revise your manuscript taking into account all reviewer comments. Please highlight all changes in the manuscript text file. At this stage we will need you to upload a copy of the manuscript in MS Word .docx or similar editable format.

*2) If you have not done so already please begin to revise your manuscript so that it conforms to our Brief Communication format instructions, available

[here](http://www.nature.com/ng/authors/article_types/index.html).

*3) Include a revised version of any required Reporting Summary:

[redacted]

We hope to receive your revised manuscript within four to eight weeks. If you cannot send it within this time, please let us know.

Sincerely,

Safia Danovi
Editor
Nature Genetics

Referee expertise:

Referee #1: synthetic lethality, HRD

Referee #2: BRCA1/2, drug discovery

Referee #3: cancer genomics

Reviewers' Comments:

Reviewer #1:

Remarks to the Author:

In the manuscript from Koh et al, the authors provide data to suggest that CX5461 is highly mutagenic. To provide some context: CX5461 has been proposed as a BRCA1/2 synthetic lethal drug and has even gone into early phase clinical trials, both as a single agent but also in combination with a PARP inhibitor. CX5461 was originally classed as a RNA PolI inhibitor and was claimed, erroneously, to be non-mutagenic (PMID: 22789538). Following this O'Neil and colleagues showed that CX5461 is indeed highly mutagenic in *C. elegans* (PMID: 32414869) and Geeleher and colleagues found that CX5461 is also a TOP2B poison (PMID: 34753908). Drugs that have off-target interactions with TOP2B, are known to cause often fatal leukemia and cardiotoxicity, in some cases several years after treatment, which would normally be enough to stop the clinical development of something like CX5461 in its tracks. Because of this, the potential safety concerns of CX5461 have already been highlighted – the Geeleher et al paper is a good example of this (PMID: 34753908).

In this manuscript, Koh et al show that not only is CX5461 mutagenic in human cells but that its mutagenic consequences appear to be uncoupled from its ability to synthetic lethal kill BRCA1 or BRCA2 cells – the prediction from this uncoupling of mutagenesis and the synthetic lethality is that CX5461 has the potential to mutate normal cells, cause secondary malignancies or other tissue damage (see comments about cardia damage above).

Overall, the experiments are well conducted. I have a few comments that I hope might add to the manuscript:

1. From the perspective of cancer treatments, the extent of mutagenesis (and therefore risk of deleterious effects) is of course relative to the therapeutic benefit of an agent, and its superiority or inferiority to what else is available to the patient. In BRCA1/2 mutant cancer, platinum and PARPi are now the standards of care and so a comparison to these agents is key. There is a comparison to the mutagenic effects of platinum salt late in the manuscript, but I did not see a comparison to PARPi which many might expect as the obvious comparison.
2. From a risk perspective, the mutagenic qualities of a drug are often balanced out by the therapeutic effect. For example, when used at high concentrations, over long periods in aged, leukemia prone, mice, PARP inhibitors induce secondary malignancies. Luckily the dosing of PARPi used in humans to get a therapeutic effect do not come close to the conditions used in mice. With this in mind, one key detail I was looking for but could not find was the selective pressure used in the in vitro mutagenesis experiments. If one is going to posit that one drug is more or less mutagenic than another, I think one has to use dosing of the two agents that gives an almost identical therapeutic effect in a BRCA isogenic system. So, were the mutagenesis experiments using CX5461 and platinum salts carried out at concentrations that reduced the population size by the same extent? What is the mutagenic frequency when PARPi are used at concentrations that reduce the population size by the same extent? If similar selective pressures were not used, it is difficult to see this as being a fair test.
3. The real impact of this work would be to highlight, with data, the real risk that could arise from the

continuation of trials using CX5461. However, to reach this impact, I can see that those who would still move forward with CX5461 trials could, with reason, argue that what is seen in this manuscript is what happens in two, quite artificial, systems, RPE1 and HAP1 cells. I can understand the rationale for using these as discovery tools, but they neither reflect the genomic complexity or presence of endogenous BRCA1/2 mutations seen in human cancers nor reflect the p53 activity and other anti-tumorigenic or anti-mutagenic mechanisms found in normal tissue. For these reasons, the remaining doubt in this area would only be removed by assessing the mutagenic effects of CX5461 in human tumor cell lines or PDOs with endogenous BRCA1/2 mutations, in normal tissue organoids and in immunocompetent mice, where even a 30 day treatment is likely to result in mutagenesis. I appreciate that for these latter models, daughter clones could not be isolated and easily analysed in the same way as per the in vitro experiments, but some assessment of the specific mutational patterns identified from the in vitro experiments could be made.

4. I appreciate that by assessing mutagenic signatures in cells exposed to drugs such as CX5461, there is the real potential to identify mechanisms of action that might not be predicted by other means. I think however that it might help to highlight that the mutagenic patterns seen in experiments using CX5461 raise mechanistic hypotheses but do not test them or add support to pre-existing mechanistic hypotheses, such as the TOP2 poison hypothesis framed by others. For example, the hypothesis about nucleosome occupancy and AT rich sequences is consistent with the data but not proven by the data provided. To mechanistically test this hypothesis, one would need to make synthetic DNA sequences which are nucleosome bound and expose these to CX5461 in the presence of TOP2. Much easier to do would be to test whether CX5461 does indeed cause the formation of bulky lesions at AT rich regions. I think these sorts of experiments are beyond a brief report, but some acknowledgement that what is reported is hypothesis-generating and not confirmatory, might be more appropriate.

Reviewer #2:
Remarks to the Author:
Koh et al. NG 2023

Previous studies have reported that CX-5461 treatment selectively kills Brca deficient tumour cells when compared to wild type controls. The current study confirms the synthetic lethal effect of CX-5461 in Brca deficient cells, thus substantiating the rationale for entering clinical trials in selected patients. Perhaps surprisingly, the mutagenic potential of CX-5461 has not been reported. This was evaluated by Koh and colleagues, who are experts in mutagenic profiling and mutation signatures. In a series of well controlled studies, the authors used clinically relevant or acute dosing schedules to compare the mutagenic potential of CX-5461 against related mechanisms (etoposide, pyridostatin), chemotherapeutics (cisplatin) and a known environmental mutagen (benzo(a)pyrene). The results unambiguously demonstrate CX-5461 is a highly dangerous mutagen, with a specific mutation signature, capable of inducing extensive mutagenic change even with a single acute dose. Critically, mutagenesis by CX-5461 is agnostic of genotype and is seen at comparable levels in control and Brca deficient backgrounds and in different cell types. Critically, the mutagenic burden induced by CX-5461 is considerably higher than all other mutagens/carcinogens tested. As such, CX-5461 can invoke synthetic lethality in Brca deficient tumours, but also induces extremely high levels of mutagenesis in the surviving normal cells. The ability of this agent to induce collateral damage in non-tumour tissues is staggering with the potential to substantially increase future cancer risk. The clinical implications of

this study are enormous, and it is imperative that this work is published without delay to ensure that clinical trials of CX-5461 are halted with immediate effect.

Reviewer #3:

Remarks to the Author:

Koh and colleagues report an extraordinary mutational burden and unique mutational signatures associated with the experimental compound CX-5461, currently being investigated in human clinical trials for treating BRCA1 / BRCA2 mutant cancers. They use a powerful system combining whole genome sequencing of clones and/or duplex sequencing of bulk populations across several drug-treated cell line models.

Their results indicate that not only BRCA1/2 deficient but normal cells may accumulate CX-5461 driven mutational damage following CX-5461 therapy. The authors claim that the mutational damage caused by CX-5461 substantially exceeds that of other DNA damaging drugs, including cisplatin, etoposide, and pyridostatin. Through elegant analyses, the authors show that G4 quadruplexes are depleted in CX-5461-driven mutational damage, while linker regions in between nucleosomes are enriched. CX-5461 driven mutations additionally show transcriptional strand asymmetry consistent with bulky adduct formation. They are also enriched in late replicating regions, though curiously do not show replication strand asymmetry.

These are fascinating results and convincing for the presence of a novel mutational process driven by CX-5461. The analyses also raise interesting questions about mechanisms of DNA damage induced by CX-5461, which has been previously proposed to target TOP2B, RNA-POL1, and G-quadruplexes. Provocatively, the authors suggest that imminent / ongoing CX-5461 trials should be halted on the basis of their results.

Comments:

- The CX-5461 doses used in the study (250nM in Fig 1 analyses, 50-200 nM) seem a bit higher than those used in previous synthetic lethality studies. For example Xu et al 2017 showed 1-10 nanomolar IC50s for BRCA2 deficient, and 10-50 nM IC50 for BRCA proficient HCT116 cells. Interestingly, the RPE1 model seems more resistant to CX-5461, for both BRCA1 wild type and mutant (Extended Data Fig 1). Can the authors comment on the relevance of these tested doses to the in vivo setting ?
- Related to above, the authors show some dosage dependence of mutagenicity of CX-5461. For example the impact of 50 nM CX-5461 on hiPSC cells (Extended Data Fig. 3b) is more modest, both relative to control and other treatments (eg BaP). This suggests that there may be a window of lower (and possibly therapeutic) doses that can be tolerated.
- It is interesting that CX-5461 causes a similar mutational burden in BRCA1/2 proficient vs deficient cells while being more lethal in the latter. This suggests that the selective lethality is independent from mutation (at least substitutions and indels). Have the authors looked at copy number or structural variants? Previous data showed γ H2ax foci eg in U2OS cells at similar dosages (100-200 nM) as those applied here.
- Can conclusions be drawn about drug mechanisms, since notably CX-5461 seems distinct from other

G4 quadruplex stabilizers (PDS) and TOP2 poisons (ETO)?

Minor comments

- Can the authors better explain Extended Data Fig. 1a? I did not understand these analyses. Cosine similarity to what? What are the bootstraps of? What are the green, pink, and blue bars?
- Some other panels could benefit from legends, axes labels, or more details in captions, for example Figure 2a, Figure 2d, right panel.
- How do the authors explain the correlation with replication timing but the absence of replicative strand asymmetry? It is a bit surprising since from previous studies the CX-5461 associated DNA damage is thought to be replication dependent. They refer to a supplementary table but it may be helpful to visualize this analysis.
- The results are exciting and possibly very timely, but perhaps some of the superlative language (alarming, dramatic, extremely, striking) can be toned down.

Author Rebuttal to Initial comments

Referee expertise:

Referee #1: synthetic lethality, HRD

Referee #2: BRCA1/2, drug discovery

Referee #3: cancer genomics

Reviewers' Comments:

Reviewer #1:

Remarks to the Author:

In the manuscript from Koh et al, the authors provide data to suggest that CX5461 is highly mutagenic. To provide some context: CX5461 has been proposed as a BRCA1/2 synthetic lethal drug and has even gone into early phase clinical trials, both as a single agent but also in combination with a PARP inhibitor. CX5461 was originally classed as a RNA PolI inhibitor and was claimed, erroneously, to be non-mutagenic (PMID: 22789538). Following this O'Neil and colleagues showed that CX5461 is indeed highly mutagenic in *C. elegans* (PMID: 32414869) and Geeleher and colleagues found that CX5461 is also a TOP2B poison (PMID: 34753908). Drugs that have off-target interactions with TOP2B, are known to cause often fatal leukemia and cardiotoxicity, in some cases several years after treatment, which would

normally be enough to stop the clinical development of something like CX5461 in its tracks. Because of this, the potential safety concerns of CX5461 have already been highlighted – the Geeleher et al paper is a good example of this (PMID: 34753908).

In this manuscript, Koh et al show that not only is CX5461 mutagenic in human cells but that its mutagenic consequences appear to be uncoupled from its ability to synthetic lethal kill BRCA1 or BRCA2 cells – the prediction from this uncoupling of mutagenesis and the synthetic lethality is that CX5461 has the potential to mutate normal cells, cause secondary malignancies or other tissue damage (see comments about cardia damage above).

Overall, the experiments are well conducted. I have a few comments that I hope might add to the manuscript:

Thank you for the kind words overall. We address your points below.

1. From the perspective of cancer treatments, the extent of mutagenesis (and therefore risk of deleterious effects) is of course relative to the therapeutic benefit of an agent, and its superiority or inferiority to what else is available to the patient. In BRCA1/2 mutant cancer, platinum and PARPi are now the standards of care and so a comparison to these agents is key. There is a comparison to the mutagenic effects of platinum salt late in the manuscript, but I did not see a comparison to PARPi which many might expect as the obvious comparison.

Thank you for raising this point.

1. Our previous work (PMID: 30982602, **Figure A** below) where we have compared and contrasted many different chemotherapeutic agents in an isogenic cellular model, has shown that PARPi Olaparib does not generate a signature. If exposure to a compound does not cause mutagenesis, then it is not a helpful comparator, and we did not use it in the paper because we knew that it caused no increase in mutation burden. We felt that it would make CX-5461 mutagenesis seem even more inflated. Quantitatively, no mutagenicity index could be calculated either.

Figure A from Zou et al, Cell 2019. Thirteen families of environmental agents are shown in different colours. For each agent, where treatment dose was selected based on IC50s and markers of DNA damage were measured, there are multiple whole genome sequenced subclones. Asterisks indicate treatments that have a significantly different mutation burden (and thus mutational signature), when compared to a cohort of fifteen different controls (far right in grey). Red arrow highlights PARPi treatment where no increase in mutation burden and no signatures were seen.

2. We have other in-house unpublished data that confirm that PARPi treatment does not induce mutagenesis in an RPE1 cells:

Sample	SBS count	Indel count	treatment
RPE1_WT_U1	3476	162	DMSO
RPE1_WT_U2	2563	154	DMSO
RPE1_WT_OL1	2775	168	Olaparib
RPE1_WT_OL2	2688	125	Olaparib
RPE1_WT_OL3	3854	175	Olaparib
RPE1_WT_OL4	3238	200	Olaparib

However, we agree with the reviewer that some general readers may seek PARPi as a direct comparison. Thus, we have now added in line 122 “we contrasted CX-5461’s mutagenicity to cisplatin and PARP inhibitor, Olaparib – alternative therapeutic agents used in *BRCA1/BRCA2*-deficient patients. While PARPi does not generate mutational signatures, cisplatin produces substitution, double substitution, and indel mutational signatures¹⁶.”

2. From a risk perspective, the mutagenic qualities of a drug are often balanced out by the therapeutic effect. For example, when used at high concentrations, over long periods in aged, leukemia prone, mice, PARP inhibitors induce secondary malignancies. Luckily the dosing of PARPi used in humans to get a therapeutic effect do not come close to the conditions used in mice. With this in mind, one key detail I was looking for but could not find was the selective pressure used in the in vitro mutagenesis experiments. If one is going to posit that one drug is more or less mutagenic than another, I think one has to use dosing of the two agents that gives an almost identical therapeutic effect in a BRCA isogenic system. So, were the mutagenesis experiments using CX5461 and platinum salts carried out at concentrations that reduced the population size by the same extent? What is the mutagenic frequency when PARPi are used at concentrations that reduce the population size by the same extent? If similar selective pressures were not used, it is difficult see this as being a fair test.

Indeed, treatment concentrations were based on the IC50s of respective agents.

To clarify, we have now added in the Methods section starting from line 373, “Cells were treated with each compound at a concentration that results in 50%–70% cytotoxicity, in parallel with cells treated with DMSO solvent control. Drug exposure frequencies, dosages, and duration are detailed in Supplemental Table 1.”

3. The real impact of this work would be to highlight, with data, the real risk that could arise from the continuation of trials using CX5461. However, to reach this impact, I can see that those who would still move forward with CX5461 trials could, with reason, argue that what is seen in this manuscript is what happens in two, quite artificial, systems, RPE1 and HAP1 cells. I can understand the rationale for using these as discovery tools, but they neither reflect the genomic complexity or presence of endogenous BRCA1/2 mutations seen in human cancers nor reflect the p53 activity and other anti-tumorigenic or anti-mutagenic mechanisms found in normal tissue. For these reasons, the remaining doubt in this area would only be removed by assessing the mutagenic effects of CX5461 are in human tumor cell lines or PDOs with endogenous BRCA1/2 mutations, in normal tissue organoids and in immunocompetent mice,

where even a 30 day treatment is likely to result in mutagenesis. I appreciate that for these latter models, daughter clones could not be isolated and easily analysed in the same way as per the *in vitro* experiments, but some assessment of the specific mutational patterns identified from the *in vitro* experiments could be made.

Thank you for the comment.

First, we feel that we have demonstrated how potently CX-5461 induces signatures in a longer exposure of 48 hours right down to a very acute exposure of 2hr, covering doses from 50 nM to 250 nM (20-30 times lower than reported max. plasma concentrations in patients administered the drug, PMID: 35750695). Mutagenesis was evident even with the smallest 2hr low dose challenge in a single-molecule read-outs through duplex sequencing. This indicates that any exposure of any cell to even the shortest duration could induce DNA damage.

Second, we have used multiple lines in keeping with the suggestions from the reviewer:

1. HAP1 that we used for the study is a cancer cell line derived from a chronic myeloid leukaemia patient with dysfunctional *TP53*.
2. RPE-1 that we used is a normal, non-transformed hTERT immortalised retinal pigmented epithelial cell line, also with *TP53* dysfunction.
3. And finally, a human induced pluripotent stem cell model, where *TP53* is functional.

While we agree with the reviewer that having *in vivo* data/model would be ideal, prior to submission, we tried to source in human *in vivo* data from current clinical trials, however there is no available material as the vast majority of samples that are obtained in clinical trials are for stratification purposes and are pre-treatment. It would be unethical to now ask for *in vivo* experiments to be conducted in humans given the data we presented here. We also approached authors of manuscripts that had worked on mouse data, but they have sadly not retained any tissue from the CX-5461-treated mice. To do new experiments would incur a huge delay, would raise questions regarding the ethics of delaying this information reaching the wider community, and be beyond the scope of a brief communication.

4. I appreciate that by assessing mutagenic signatures in cells exposed to drugs such as CX5461, there is the real potential to identify mechanisms of action that might not be predicted by other means. I think however that it might help to highlight that the mutagenic patterns seen in experiments using CX5461 raise mechanistic hypotheses but do not test them or add support to pre-existing mechanistic

hypotheses, such as the TOP2 poison hypothesis framed by others. For example, the hypothesis about nucleosome occupancy and AT rich sequences is consistent with the data but not proven by the data provided. To mechanistically test this hypothesis, one would need to make synthetic DNA sequences which are nucleosome bound and expose these to CX5461 in the presence of TOP2. Much easier to do would be to test whether CX5461 does indeed cause the formation of bulky lesions at AT rich regions. I think these sorts of experiments are beyond a brief report, but some acknowledgement that what is reported is hypothesis-generating and not confirmatory, might be more appropriate.

Indeed, thank you for the suggestion as a follow-up piece. We have included this in the manuscript where we acknowledge that mechanisms underpinning cytotoxicity are likely distinct from mechanisms underpinning mutagenicity.

Line 86:

“Taken together, our analyses suggest that while the cytotoxic effects of CX-5461 may be driven through TOP2 poisoning caused by G4 stabilisation, its mutagenic effects likely stem from alternative mechanisms – plausibly bulky, DNA-deforming adducts occurring at exposed, AT-rich genomic regions, in a sudden and catastrophic manner, accounting for the conspicuous topographical distributions noted above.”

Reviewer #2:

Remarks to the Author:

Koh et al. NG 2023

Previous studies have reported that CX-5461 treatment selectively kills Brca deficient tumour cells when compared to wild type controls. The current study confirms the synthetic lethal effect of CX-5461 in Brca deficient cells, thus substantiating the rationale for entering clinical trials in selected patients. Perhaps surprisingly, the mutagenic potential of CX-5461 has not been reported. This was evaluated by Koh and colleagues, who are experts in mutagenic profiling and mutation signatures. In a series of well controlled studies, the authors used clinically relevant or acute dosing schedules to compare the mutagenic potential of CX-5461 against related mechanisms (etoposide, pyridostatin), chemotherapeutics (cisplatin) and a known environmental mutagen (benzo(a)pyrene). The results unambiguously demonstrate CX-5461 is a highly dangerous mutagen, with a specific mutation signature, capable of inducing extensive mutagenic change even with a single acute dose. Critically, mutagenesis by CX-5461 is agnostic of genotype and is seen at comparable levels in control and Brca deficient backgrounds and in different cell types. Critically, the mutagenic burden induced by CX-5461 is considerably higher than all other mutagens/carcinogens tested. As such, CX-5461 can invoke synthetic lethality in Brca deficient tumours, but also induces extremely high levels of mutagenesis in the surviving normal cells. The ability of this agent to induce collateral damage in non-tumour tissues is staggering with the potential to substantially increase future cancer risk. The clinical implications of this study are enormous, and it is

imperative that this work is published without delay to ensure that clinical trials of CX-5461 are halted with immediate effect.

Thank you for the positive feedback and understanding of urgency. It is hugely appreciated!

Reviewer #3:

Remarks to the Author:

Koh and colleagues report an extraordinary mutational burden and unique mutational signatures associated with the experimental compound CX-5461, currently being investigated in human clinical trials for treating BRCA1 / BRCA2 mutant cancers. They use a powerful system combining whole genome sequencing of clones and/or duplex sequencing of bulk populations across several drug-treated cell line models.

Their results indicate that not only BRCA1/2 deficient but normal cells may accumulate CX-5461 driven mutational damage following CX-5461 therapy. The authors claim that the mutational damage caused by CX-5461 substantially exceeds that of other DNA damaging drugs, including cisplatin, etoposide, and pyridostatin. Through elegant analyses, the authors show that G4 quadruplexes are depleted in CX-5461-driven mutational damage, while linker regions in between nucleosomes are enriched. CX-5461 driven mutations additionally show transcriptional strand asymmetry consistent with bulky adduct formation. They are also enriched in late replicating regions, though curiously do not show replication strand asymmetry.

These are fascinating results and convincing for the presence of a novel mutational process driven by CX-5461. The analyses also raise interesting questions about mechanisms of DNA damage induced by CX-5461, which has been previously proposed to target TOP2B, RNA-POL1, and G-quadruplexes. Provocatively, the authors suggest that imminent / ongoing CX-5461 trials should be halted on the basis of their results.

Thank you for the supportive review.

Comments:

1. The CX-5461 doses used in the study (250nM in Fig 1 analyses, 50-200 nM) seem a bit higher than those used in previous synthetic lethality studies. For example, Xu et al 2017 showed 1-10 nanomolar IC50s for BRCA2 deficient, and 10-50 nM IC50 for BRCA proficient HCT116 cells. Interestingly, the RPE1 model seems more resistant to CX-5461, for both BRCA1 wild type and mutant (Extended Data Fig 1). Can the authors comment on the relevance of the these tested doses to the in vivo setting?

Thank you for raising this interesting point.

In Xu et al 2017 (PMID: 28211448), the IC50s reported for both *BRCA2*-null DLD-1 and HCT116 cells are indeed in the lower nanomolar ranges. However, both cell line models are not “clean” *BRCA*-deficient models – they both have a compound phenotype of mismatch repair deficiency (MMRd) in addition to being *BRCA*-deficient.

- HCT-116 is known to carry biallelic *MLH1* mutations
- DLD-1 carries biallelic *MSH6* mutations, with also a reported mutation in *POLD1*.

Thus, the additional MMRd status of these models likely amplifies the impact of CX-5461 toxicity (PMID: 32414869).

In other *BRCA*-deficient human cellular models, however (Supplementary Table 4 of PMID: 32414869), where alternative cellular models were used, HCC1806 and HCC1937, the IC50s were consistently in the range between 55 – 200 nM, and similar to ours.

Finally, addressing the point about our *in-vitro* tested concentrations to those observed *in vivo*; CX-5461 is administered by IV infusion with average day one maximum plasma concentration of ~2-3uM (PMID: 35750695). This is 20-30 times higher than what we have used in our *in vitro* models (0.05-0.2uM). Given the magnitude of the mutagenesis seen here and the reported slow elimination half-life ($t_{1/2}$: 61.5 ± 15.5 h) of CX-5461, we feel that the concentrations we have selected are justified and pharmacologically relevant.

2. Related to above, the authors show some dosage dependence of mutagenicity of CX-5461. For example the impact of 50 nM CX-5461 on hiPSC cells (Extended Data Fig. 3b) is more modest, both relative to control and other treatments (eg BaP). This suggests that there may be a window of lower (and possibly therapeutic) doses that can be tolerated.

Thank you. See final paragraph of response to point #1 for interpreting this in the context of an *in vivo* setting.

3. It is interesting that CX-5461 causes a similar mutational burden in *BRCA1/2* proficient vs deficient cells while being more lethal in the latter. This suggest that the selective lethality is independent from mutation (at least substitutions and indels). Have the authors looked at copy number or structural

variants? Previous data showed γ H2ax foci eg in U2OS cells at similar dosages (100-200 nM) as those applied here.

Thank you.

SV yields in experimental subclones were low (avg. SV \sim 9 per sample, provided in Supplementary Table S2). CX-5461-treated subclones did show a slight increase in SV counts compared to their untreated counterparts (mean SV count: 11.5 vs. 7 in treated vs. untreated control cells), but because of such low numbers, we chose to conservatively focus on the profound SBS and indel phenotypes. The same thing was observed for the chromosomal level copy number variations. There seems to be a very small difference between clones treated and not treated with CX-5461 (see graph below).

However, given that the reviewer has kindly raised the point, we have now added “Although CX-5461-treated subclones did show a slight increase in SV counts and some but not very consistent chromosomal copy number aberrations compared to their untreated counterparts, the counts were too low and insufficiently powered to draw any conclusions.” to the Method section line 426. Copy number data are also provided in Supplementary Table 2.

Although previous publications have demonstrated successful induction of γ H2Ax foci in experimental models exposed to CX-5461, we have shown in our previous work (PMID: 30982602) that successful DNA damage induction does not always correlate with mutagenic outcome; the reverse is also true. *E.g.*, formaldehyde treatment does not induce detectable DDR signalling in human iPSC cells but is associated with a mutation pattern, whereas acetaldehyde and acrylamide are able to elicit DDR, but do not produce detectable mutation patterns. Thus, DDR induction does not necessarily predict mutagenesis.

4. Can conclusions be drawn about drug mechanisms, since notably CX-5461 seems distinct from other G4 quadruplex stabilizers (PDS) and TOP2 poisons (ETO)?

From our work, we cannot provide definitive insights into the exact molecular mechanisms underpinning cytotoxicity.

However, our analyses do suggest that:

- CX-5461 activity may be driven through G4 stabilisation
- its mutagenic effects (especially for SBS) are likely effected through alternative mechanisms.

Collectively, the strong transcriptional strand bias of the SBS, and the hypothesis about nucleosome occupancy and enrichment at AT-rich sequences are all consistent with published literature and alluded to in the manuscript.

To mechanistically test this hypothesis, one would need to make synthetic DNA sequences which are nucleosome bound and expose these to CX5461 in the presence of TOP2 as suggested by Reviewer 1. We feel and Reviewer 1 also states that these sorts of experiments are beyond a brief report.

Minor comments

5. Can the authors better explain Extended Data Fig. 1a? I did not understand these analyses. Cosine similarity to what? What are the bootstraps of? What are the green, pink, and blue bars?

Yes of course. Previous figure legends were trimmed down because of the word limits associated with a Brief Communication. We have now revised the legends for clarification and referred the reader to the Methods for details.

Extended Data Fig. 1a

“Distinguishing *de novo* mutational profiles of experimental subclones from vehicle-treated controls. Light green, pink, and blue error bars depict the distribution of cosine similarities between bootstrapped control subclone profiles and the aggregated control mutational profile of respective mutation types with increasing mutation counts. The x axis displays the *de novo* mutation counts for respective mutation classes. See Methods”

Under “Method” from line 406

“Mutational signature analysis of experimental samples

Experimental mutational signatures were derived using published framework based on cosine similarity, profile bootstrapping, and background subtraction^{21,25}. Briefly, we first (i) determined the background mutational signature in unedited/untreated control by aggregating the unedited and untreated subclone mutational profiles; (ii) assessed the difference/s between the mutational profiles of the edited/treated clones and the controls using cosine similarity. Specifically, we first evaluated the similarity of mutational profiles between the untreated control and each subclone. We calculated the cosine similarity between each bootstrapped control sample and the aggregated background control mutational signature from (i) (means and standard deviations). A cosine similarity close to 1.0 indicates that the mutation profile of the bootstrapped sample is near-identical to the control signature. Cosine similarities could thus be considered across a range of mutation burdens (green, pink, and blue line for substitution, double substitution, and indel respectively, in Extended Data Fig. 1a). Next, we calculated cosine similarities between edited/treated subclone profiles and control (coloured shapes in Extended Data Fig. 1a). An edit or a treatment that does not fall within the expected distribution of cosine similarities implies a mutation profile distinct from controls (that is, the perturbation generated a signature). If an edit or a treatment generates a signature, we (iii) removed background mutation profile from the mutation profile of edited/treated clones (Supplementary Table 2).”

6. Some other panels could benefit from legends, axes labels, or more details in captions, for example Figure 2a, Figure 2d, right panel.

Understood. Figure legends have now been revised for clarity for Figure 2a.
Axes labels added to Figure 2d right panel.

7. How do the authors explain the correlation with replication timing but the absence of replicative strand asymmetry? It is a bit surprising since from previous studies the CX-5461 associated DNA damage is thought to be replication dependent. They refer to a supplementary table but it may be helpful to visualize this analysis.

In the previous study (PMID: 32457376), CX-5461-induced damage (as measured by γ H2Ax) was suggested to be replication-dependent in EdU-positive HR-proficient cells, but crucially, also in both EdU+ve and EdU-ve populations of HR-deficient cells. This suggests that replication (stress) is not the only cause of CX-5461 induced damage.

Also, γ H2Ax is not a marker for base lesions in particular and is more of a generic marker of DNA damage (often assumed to be associated with strand breaks). Note that in this data, the greatest effect in terms of mutagenicity is single base substitutions (most likely caused by single base lesions). It's also useful to point out that the strand bias and replication timing analyses were done with substitution data only because indel and SV counts were too low, precluding such analyses owing to insufficient statistical power.

Having said that, we did note that our previous description of the replication timing data was not clear and indeed the reviewer has picked up a really important message that we did not intend to communicate. As seen in Figure 2c, the better way for us to express this is that observed mutation distribution tracks the expected mutations (dotted line) given the trinucleotide sequence context of the signature, suggesting that cell cycle timing does not influence how the mutations are fixed at all. This has now been edited from line 77, "Moreover, CX-5461 mutations were evidently enriched in AT-rich, open chromatin regions, unaffected by replication timing (**Fig. 2c**), befitting rapid and substantial DNA damage engendered by CX-5461, primarily at open, exposed AT-rich regions."

The replication strand bias result was included as a supplementary table instead of a plot as we're limited by the number of figures for a Nature Genetics brief communication. The image is shown below. The absence of replication strand bias does not contradict the prior suggestion that *cytotoxic* impact is replication dependent.

8. The results are exciting and possibly very timely, but perhaps some of the superlative language (alarming, dramatic, extremely, striking) can be toned down.

Understood. We have reduced these emotive words in the text:

- alarmingly and disturbingly (final para page 1) – line 35
- the striking (middle para page 2) – line 66

Decision Letter, first revision:

17th Oct 2023

Dear Serena,

Thank you for submitting your revised manuscript "The chemotherapeutic CX-5461 is extremely mutagenic and may increase cancer risk" (NG-BC63311R). It has now been seen by the original referees and their comments are below. The reviewers find that the paper has improved in revision, and therefore we'll be happy in principle to publish it in Nature Genetics, pending minor revisions to satisfy our editorial and formatting guidelines.

There is a (very) slim chance that we can rush this through for publication this year but I can't make any promises. I will do what I can at this end but there are parts of the process that I have no control over, so please don't hold me to this! We're now performing detailed checks on your paper and will send you a formal checklist detailing our editorial and formatting requirements soon. Please do not upload the final materials and make any revisions until you receive this additional information from us. But one thing you can start doing is ensuring that all your data and code are deposited and fully available (we won't accept a paper unless this has been done). I think that we also need to be very

careful about the language in the paper, and (as suggested by one of your reviewers) ensure that both the wording and tone throughout is appropriately tempered, and fully supported by the data.

Sincerely,

Safia Danovi
Editor
Nature Genetics

Reviewer #1 (Remarks to the Author):

i think that on balance, the authors have provided the information that is needed to move forward. For me the critical issue was clarification that the selective pressure used in the mutagenesis expts was comparable to that in their prior expts using parp inhibitor etc.. This for me crystallises the strong possibility that when used at a dose that might cause a BRCA synthetic lethality, the potential for toxicity to normal tissue caused by genotoxicity or secondary malignancies caused by mutagenesis are high.

I think it would have been nice to have the mutagenic profiles confirmed in more clinically relevant model systems but this should be balanced against the urgency in reporting this data (and i dont think this should preclude publication). For example, if I were running a trial with this agent, upon seeing this data I would be assessing the mutagenic effects on the drug on lymphocytes from what i imagine (hope) are routine bloods that have been collected before, on and after treatment. That would be probably the best test of the overarching hypothesis.

Reviewer #2 (Remarks to the Author):

The authors have provided further evidence to support their original conclusions in response to the 2 other reviewers. I believe they have addressed these points satisfactorily and therefore the paper should be published wit a matter of urgency.

Reviewer #3 (Remarks to the Author):

The authors have adequately addressed my comments. I congratulate them for an elegant and timely study.

Final Decision Letter:

30th Oct 2023

Dear Dr Nik-Zainal,

I am delighted to say that your manuscript "The chemotherapeutic drug CX-5461 is a potent mutagen in cultured human cells" has been accepted for publication in an upcoming issue of Nature Genetics.

Your paper will be published online after we receive your corrections and will appear in print in the next available issue. You can find out your date of online publication by contacting the Nature Press Office (press@nature.com) after sending your e-proof corrections. Now is the time to inform your Public Relations or Press Office about your paper, as they might be interested in promoting its publication. This will allow them time to prepare an accurate and satisfactory press release. Include your manuscript tracking number (NG-BC63311R1) and the name of the journal, which they will need when they contact our Press Office.

Please note that *Nature Genetics* is a Transformative Journal (TJ). Authors may publish their research with us through the traditional subscription access route or make their paper immediately open access through payment of an article-processing charge (APC). Authors will not be required to make a final decision about access to their article until it has been accepted. [Find out more about Transformative Journals](https://www.springernature.com/gp/open-research/transformative-journals)

Authors may need to take specific actions to achieve [compliance](https://www.springernature.com/gp/open-research/funding/policy-compliance-faqs) with funder and institutional open access mandates. If your research is supported by a funder that requires immediate open access (e.g. according to [Plan S principles](https://www.springernature.com/gp/open-research/plan-s-compliance)) then you should select the gold OA route, and we will direct you to the compliant route where possible. For authors selecting the subscription publication route, the journal's standard licensing terms will need to be accepted, including <https://www.nature.com/nature-portfolio/editorial-policies/self-archiving-and-license-to-publish>. Those licensing terms will supersede any other terms that the author or any third party may assert apply to any version of the manuscript.

If you have not already done so, we invite you to upload the step-by-step protocols used in this manuscript to the Protocols Exchange, part of our on-line web resource, natureprotocols.com. If you complete the upload by the time you receive your manuscript proofs, we can insert links in your article that lead directly to the protocol details. Your protocol will be made freely available upon publication of your paper. By participating in natureprotocols.com, you are enabling researchers to more readily reproduce or adapt the methodology you use. [Natureprotocols.com](http://natureprotocols.com) is fully searchable, providing your protocols and paper with increased utility and visibility. Please submit your protocol to <https://protocolexchange.researchsquare.com/>. After entering your [nature.com](http://www.nature.com) username and password you will need to enter your manuscript number (NG-BC63311R1). Further information can be found at <https://www.nature.com/nature-portfolio/editorial-policies/reporting-standards#protocols>

Sincerely,

Safia Danovi
Editor
Nature Genetics